# Dazzling Evaluation of the Impact of a High-Repetition-Rate CO_2_ Pulsed Laser on Infrared Imaging Systems

**DOI:** 10.3390/s24061827

**Published:** 2024-03-12

**Authors:** Hanyu Zheng, Yunzhe Wang, Yang Liu, Tao Sun, Junfeng Shao

**Affiliations:** 1State Key Laboratory of Laser Interaction with Matter, Changchun Institute of Optics, Fine Mechanics and Physics, Chinese Academy of Sciences, Changchun 130033, China; zhenghanyu22@mails.ucas.ac.cn (H.Z.); wangyunzhe20@mails.ucas.ac.cn (Y.W.); liuyangdk@ciomp.ac.cn (Y.L.); suntao@ciomp.ac.cn (T.S.); 2University of Chinese Academy of Sciences, Beijing 100049, China

**Keywords:** laser dazzling, edge extraction, contour curvature, target detection

## Abstract

This article utilizes the Canny edge extraction algorithm based on contour curvature and the cross-correlation template matching algorithm to extensively study the impact of a high-repetition-rate CO_2_ pulsed laser on the target extraction and tracking performance of an infrared imaging detector. It establishes a quantified dazzling pattern for lasers on infrared imaging systems. By conducting laser dazzling and damage experiments, a detailed analysis of the normalized correlation between the target and the dazzling images is performed to quantitatively describe the laser dazzling effects. Simultaneously, an evaluation system, including target distance and laser power evaluation factors, is established to determine the dazzling level and whether the target is recognizable. The research results reveal that the laser power and target position are crucial factors affecting the detection performance of infrared imaging detector systems under laser dazzling. Different laser powers are required to successfully interfere with the recognition algorithm of the infrared imaging detector at different distances. And laser dazzling produces a considerable quantity of false edge information, which seriously affects the performance of the pattern recognition algorithm. In laser damage experiments, the detector experienced functional damage, with a quarter of the image displaying as completely black. The energy density threshold required for the functional damage of the detector is approximately 3 J/cm^2^. The dazzling assessment conclusions also apply to the evaluation of the damage results. Finally, the proposed evaluation formula aligns with the experimental results, objectively reflecting the actual impact of laser dazzling on the target extraction and the tracking performance of infrared imaging systems. This study provides an in-depth and accurate analysis for understanding the influence of lasers on the performance of infrared imaging detectors.

## 1. Introduction

The role and status of electro-optical-guided detectors have been increasingly prominent in modern high-tech warfare, especially in the striking of high-value targets through the remarkable cost-effectiveness and combat power demonstrated by infrared imaging detectors. The suppression of laser dazzling is considered an effective means to counteract them. This method of dazzling saturates or damages the infrared imaging devices on these detectors using lasers, rendering them incapable of target identification and tracking.

The irradiation effects of lasers on detectors can be classified as photoelectric effects and material thermal effects [1,2,3]. As the energy of laser dazzling continues to increase, saturation, saturation crosstalk, and over-saturation phenomena can be observed in the images [4,5,6]. Further increases in the laser power result in point, line, surface, functional damage, and blinding damage effects on the images [7,8,9,10]. When the detector experiences functional damage, a quarter of the image area becomes blind, and this damage is irreversible. Therefore, a thorough analysis of the laser dazzling effect on the image at this point, especially its impact on the target detection performance, is of crucial military significance and practical importance.

Currently, scholars both domestically and internationally have conducted research on the evaluation of laser dazzling effects from various perspectives, including the impact of laser dazzling on the target detection performance. From the perspective of the impact on target detection performance using laser dazzling assessment methods, Durécu et al. proposed an algorithm based on edge detection, using correlation and a Fourier descriptor as two evaluation indicators to quantitatively assess the impact of laser dazzling on the pattern recognition algorithm performance [11]. Hueber et al. used pattern recognition to assess the impact of laser dazzling on image edge information. They applied the Canny algorithm to the original and dazzling images for image processing and quantified the degree of loss using the correlation coefficient of each target image’s edge information [12]. Jie et al. analyzed the jamming effect of a synthetic fiber laser on an airborne defense system and evaluated the four-image guidance system and image guidance system based on the synthesized fiber laser [13]. Gao et al. systematically analyzed the impact of laser dazzling on target detection algorithms from the perspective of machine learning. By calculating the occlusion rate of the target, the correlation between the target and the target template, and further analyzing the changes in the output features of the images before and after laser dazzling in convolutional networks, they divided the effective dazzling area of the dazzling spot [14]. Fengling et al. used an improved method of image correlation to assess laser dazzling effects. By determining the correlation threshold through human eyes, they accurately evaluated dazzling images under different bands and laser powers, and this method demonstrated a certain robustness [15]. Gareth et al. used a scale-invariant template matching algorithm to establish an architecture and employed artificial intelligence algorithms to quantify the effects. They compared the similarity of images before and after laser dazzling and assessed the impact of the laser on the tracking performance of infrared imaging guidance heads [16].

This article utilizes the Canny edge detection algorithm based on contour curvature and the cross-correlation template matching algorithm to systematically study the evaluation of laser dazzling effects on an infrared detector at both distant and close ranges through experiments on laser dazzling and laser damage effects. It analyzes the degree of image distortion before and after laser dazzling when functional damage occurs in the infrared imaging system, providing quantitative practical guidelines and methods for evaluating laser dazzling effects. During the dazzling process, the algorithms examine the correlation under different laser powers and target positions and provide a set of image quality evaluation functions and thresholds to determine the dazzling level and its recognizability. Specifically, by calculating the norm function N and comparing it with the preset threshold N_0_, if N is lower than N_0_, the dazzling is determined to be effective.

## 2. Canny Edge Extraction Algorithm Based on Contour Curvature

Edge extraction is the most fundamental feature of an image, and preprocessing the image with edge extraction before applying template matching algorithms on interfered images accelerates image processing speed and achieves real-time processing effects [17]. Edges are collections of pixels with abrupt changes in pixel grayscale values, and image edges reflect most of the image information [18]. Compared to the existing edge extraction algorithms, such as Robert, Sobel, and Prewitt, the Canny edge extraction algorithm is considered to be more insensitive to noise and has better anti-dazzling capabilities. This is highly beneficial for the accuracy and speed of subsequent pattern recognition algorithms [19]. In laser active imaging, noise mainly consists of speckle noise, CCD noise, and environmental background noise. Conventional edge extraction operators are not effective in extracting contour information from the original image [20,21]. Therefore, a Canny edge extraction algorithm based on contour curvature is proposed. Firstly, Gaussian filtering is applied to smooth the original image, effectively eliminating some noise in the image. Secondly, differential processing is used to calculate the gradient magnitude and direction of selected edge points, obtaining edge information. Non-maximum suppression is applied to refine the edges. Finally, contour curvature is utilized to classify the edge information. By connecting strong edges and neighboring weak edges, the final edge detection and image segmentation are completed.

### 2.1. The Gaussian Low-Pass Filter Smooths the Original Image

The transfer function of Gaussian smoothing is defined as $h\left(x,y,\sigma \right)=\exp\left(-\frac{x^{2}+y^{2}}{2\sigma^{2}}\right)/2\pi \sigma^{2}$, where x and y represent the pixel points in the horizontal and vertical directions of the image, respectively, and σ is a dimensionless coefficient. The Gaussian smoothing of the original image $f\left(x,y\right)$ results in the processed image $g\left(x,y\right)=h\left(x,y,\sigma \right)*f\left(x,y\right)$, where ∗ denotes convolution. The function $h\left(x,y,\sigma \right)$ is transformed into a two-dimensional template for convolution with the image. By converting the Gaussian smoothing process into a two-dimensional template for convolution, noise in the image can be effectively eliminated, making the calculation of derivatives more stable. The application of smoothing filters helps to improve the robustness of edge detection algorithms, ensuring that the algorithm can accurately capture the actual edge features in the image without being affected by noise.

### 2.2. Image Edge Information Is Obtained Based on Differential Processing

Generally speaking, images can be represented and classified in both Fourier space and real space. In Fourier space, the representation of an image consists of low- to high-frequency information. By applying high-pass or low-pass filtering functions, it is possible to extract edge information or perform denoising (i.e., image smoothing). Similarly, in real space, images are composed of pixels with different amplitude values, and pixels at the edges exhibit relatively large amplitude variations. Therefore, by performing differential processing on the real-space information of an image, it is possible to enhance and highlight drastic amplitude changes in pixels, achieving the extraction of edge information. Based on the image $g\left(x,y\right)\;$obtained from the previous section after smoothing, Formula (1) is applied for differential processing in the horizontal and vertical directions (i.e., x  and y directions, respectively) of the image.
(1)$$\left\{\begin{array}{c} G_{x}=\frac{g\left(x+1,y\right)-g\left(x,y\right)+g\left(x+1,y+1\right)-g\left(x,y+1\right)}{2}\\ G_{y}=\frac{g\left(x,y+1\right)-g\left(x,y\right)+g\left(x+1,y+1\right)-g\left(x+1,y\right)}{2} \end{array}\right.$$

The amplitude information $M\left(x,y\right)$ and direction information $\theta \left(x,y\right)$ in the image after differential processing can be expressed using Formulas (2) and (3), respectively.
(2)$$M\left(x,y\right)=\sqrt{G_{x}^{2}+G_{y}^{2}}$$
(3)$$\theta \left(x,y\right)=\arctan\left(\frac{G_{x}}{G_{y}}\right)$$
where $G_{x}$ and $G_{y}$ represent the first partial derivatives in the x and y directions, respectively.

Non-maximum suppression is used to ensure the accurate extraction of edge information in the processed image. The edge information at this stage includes both genuine edge information and pseudo-edges generated by noise. To preserve the characteristic information in genuine edge information, an edge filtering method is employed. This involves selecting the edge length threshold *L_d_* and the edge strength threshold *L_s_*. If the length of a certain edge segment is less than *L_d_* or the strength of the edge is less than *L_s_*, it is considered that the edge is caused by noise. The pixels on such edges are classified as non-edge points, resulting in the generation of a candidate edge image.

### 2.3. Edge Information Is Divided Based on Contour Curvature

By applying the eleven-point curvature method to calculate the contour curvature, we can accurately measure the curvature features of an object’s outline. The calculation method for contour curvature involves analyzing eleven points on the object’s contour to obtain detailed curvature information. This curvature value can be used to assess the importance of edge information because curvature exhibits different characteristics in different parts of the contour, thereby enhancing the recognition of edge information. Contour curvature to assess the importance of edge information is defined as:(4)$$K=sign\left[\left(x_{i}-x_{i-5}\right)\left(y_{i+5}-y_{i}\right)-\left(y_{i}-y_{i-5}\right)\left(x_{i+5}-x_{i}\right)\right]\frac{V_{i1}V_{i2}}{\left|\left|V_{i1}\right|\right|\left|\left|V_{i2}\right|\right|}$$

Let $G_{i}$ be pixels, where $V_{i1}=G_{i}-G_{i-5}$,$V_{i2}=G_{i+5}-G_{i}$. This curvature value measures the degree of curvature in different parts of the contour, aiding to achieve a more accurate understanding of the shape and features of the object’s outline, especially for target detection and analysis in image processing and computer vision applications.

By setting a contour curvature threshold $L_{k}$, we can divide the edge information into a strong edge, weak edge, and non-edge. For strong edges, the value of the contour curvature K is larger. For weak edges, the curvature K value is small. Anything else is non-marginal.

The edge information function is defined as $T\left(p\right)$, and the edge information is divided into: when $T\left(p\right)=1$, the pixel is a strong edge pixel; when $T\left(p\right)=0$, the pixel is a weak edge pixel. When $T\left(p\right)=-1$, it means that the pixel is a non-edge pixel, that is, the contour curvature K does not exist. The value of $T\left(p\right)$ is
(5)$$T\left(p\right)=\left\{\begin{array}{cc} 1, & \left|K\right|\geq K_{t}\\ 0, & \left|K\right|<K_{t}\\ -1, & \left|K\right|does\;not\;exist \end{array}\right.$$

Formula (5) is utilized to iterate through the candidate edge image to classify the edge pixels. Strong edge pixels are directly labeled as edge pixels. For weak edge pixels, it is necessary to examine the surrounding pixels. If there are adjacent strong edge pixels, the weak edge pixel is labeled as an edge pixel as well. This process effectively connects the edges, forming continuous edge lines and resulting in the final edge image.

Figure 1 shows the image of the target image and its 3D intensity map after applying the improved Canny edge extraction algorithm.

## 3. Cross-Correlation Template Matching Algorithm

After processing the laser dazzling image with the Canny algorithm, this paper employed the cross-correlation template matching algorithm for image matching. The template matching algorithm first reads the original image and the template image, converting them into grayscale images. The sizes of the original image and the template image are obtained, and the range to be traversed by the two images is calculated through subtraction. An empty result matrix is created to store the correlation at each position with the template. The template image is converted into a vector, and its norm is calculated for subsequent correlation calculations. Using two nested loops, each position in the original image is traversed, starting from the top-left corner, and the correlation is calculated sequentially. At each position, a sub-matrix of the same size as the template image is extracted from the original image and converted into a vector. The dot product of the sub-matrix from the original image and the vector from the template image is calculated, and the result is divided by the product of their norms to obtain the normalized correlation $N\left(x,y\right)$.
(6)$$N\left(x,y\right)=\frac{\Sigma \left(\left(G(x,y)-\mu G)(S(x,y)-\mu S\right)\right)}{\sqrt{\Sigma {\left(G\left(x,y\right)-\mu G\right)}^{2}*\Sigma {\left(S\left(x,y\right)-\mu S\right)}^{2}}}$$
where G is the template image processed by the Canny algorithm, S is the laser dazzling image, and μG and μS are the image average of the above two, respectively. The normalized relevance is stored in the corresponding position of the result matrix, and then the position with the greatest relevance is found in the result matrix, that is, the matched position. Matching positions can be shown by drawing marks over the original image.

## 4. Experimental Setup

This study employed a small TEA high-repetition-rate CO_2_ pulsed laser for both experiments (pulse width: 150 ns; wavelength: 10.6 μm; laser divergence angle: 5 mrad; and Gaussian beam). The detector used was a long-wave infrared uncooled camera, model PLUG417R (resolution: 400 × 300; pixel size: 17 μm; and lens aperture: 50 mm).

### 4.1. Laser Dazzling Effects’ Experiment

To conduct experiments on the dazzling effects of the 10.6 μm pulse laser on infrared imaging systems, the pulse repetition frequency used in this experiment was 100 Hz, and the irradiation time was 1 s. The dazzling distance was set at 50.8 m. The experimental setup is illustrated in Figure 2.

The pulse laser light source passes through a beam splitter and an attenuator before irradiating the surface of the infrared detector. The beam splitter is used for spectral splitting, with a small portion of weak light entering a power meter for the real-time monitoring of laser power. The majority of the laser is directed onto the detector. The attenuator controls the degree of laser energy attenuation. The detector is connected to a computer to observe and record changes in the image in real time.

The experimental procedure is as follows:(1)Construct the experimental optical path as shown in Figure 2.(2)Measurement of the target power: Placing a power meter behind the camera, test the first step without adding attenuators, calibrate the energy attenuation of the optical path with two power meters separately, and measure the initial transmittance as well as the attenuation factor of the attenuator. Divide the target power by the power measured by the spectroscope and the laser frequency to obtain the attenuation ratio of the optical path, denoted as R_1_ (R_1_ = P_Target_/P_BS_R_/f_1_); subsequently, insert the attenuators A, B, C, and D separately to measure the new attenuation ratio of the optical path after passing through a single attenuator, denoted as R_2_ (R_2_ = P_Target_/P_BS_R_/f_2_). The ratio of R_2_ to R_1_ represents the attenuation factor of a single attenuator. During the experiment, we added different combinations of attenuators; the laser power measured by the beam splitter was denoted as P and the laser repetition frequency as f_1_ = f_2_. Multiplying P by R_1_ and then by the attenuation factor of the attenuator yields the actual single-pulse energy to the target of the laser damage detector. Dividing the actual single-pulse energy to the target by the spot area provides the laser energy density, as shown in Table 1.(3)Adjust the lens to focus on the spot; with the detector lens cover on, turn on the laser to irradiate the center of the target surface of the detector. Add all attenuators to prevent damaging the detector, adjust the lens focal length to minimize the focused spot, and capture the laser spot image. This spot will be used for the subsequent calculations of the target power density.(4)Replace the attenuators according to the combination list and conduct dazzling experiments on the detector under different laser powers. The computer monitors the dazzling effects of the detector under different laser powers and collects images accordingly.

### 4.2. Laser Damage Effects’ Experiment

We conducted long-distance damage experiments in an outdoor setting, and the experimental schematic is shown in Figure 3. The distance between the laser and the damage target was approximately 2 km, requiring a beam expansion treatment for the laser to compress the laser divergence angle, reduce the diameter of the spot on the front surface of the target before reaching it, and increase the energy density on the target. To achieve this, a beam expander system was assembled for the laser in this experiment. The beam expander systems used in the experiment were the LZBE-1060-2X laser beam expander mirror from the LBTEK Company (Shenzhen, China) and the BEF05R-AG reflective beam expander. The system employed two-stage expansion, initially performing 2× coaxial expansion on the beam emitted from the laser and then undergoing 5× expansion through the off-axis expansion system, ultimately achieving 10× expansion. The theoretical compression of the laser’s far-field divergence angle is 10 times, reaching 0.5 mrad. Combining the exit spot diameter of the expansion system, the theoretical value of the spot diameter at a distance of 1.8 km is approximately 1 m. Therefore, taking advantage of the characteristic that the spot radius in front of the target surface under far-field conditions is much larger than the aperture of the camera, the power meter probe and the camera were placed parallelly and in close adjacence. The lens 3 was placed in front of the probe to receive the laser, enabling the simultaneous camera monitoring and power measurement. In this setup, lenses 1 and 2 were used as camera lenses, while lens 3 served as the focusing lens in front of the energy meter.

To ensure consistency in the irradiation time, before conducting the damage test, we covered lens 1 with a light-shielding plate. Then, we activated the camera recording function until the laser irradiation ended. We removed the light-shielding plate and inspected the imaging performance of the camera, and then ended the recording. A total of 2 camera lenses (camera lens 1 and camera lens 2) were damaged during the experiment, and the camera models were identical.

## 5. Results and Discussion

### 5.1. Experimental Results and Analysis of the Laser Dazzling Effect

To investigate the overall assessment of the laser dazzling effects, experiments were conducted using the experimental setup for laser dazzling effects shown in Figure 2, exploring the infrared laser dazzling effects on the targets at different laser power levels and distances. Figure 4 displays the positions of the laser spot and the target objects, where A represents the distance between the center of the spot and the target object. The measurement results are shown in Table 2.

To explore the overall assessment of the laser dazzling images, experiments were conducted using the aforementioned experimental setup with different laser powers and target positions. A series of partial laser dazzling images were obtained, as shown in Figure 5a,c.

The laser dazzling images obtained from the experiment were processed and recognized using the Canny algorithm based on contour curvature and the cross-correlation template matching algorithm introduced in the Section 2. The recognized results after algorithm processing show that the laser power has a significant impact on the dazzling results. When the target is at a long distance (r = 1.7 m), dazzling can be successful when the laser power P ≥ 1.5 mW. However, when the target is at a close distance (r = 1.1 m), dazzling can only be successful when the laser power P ≥ 0.4 mW, causing the failure of the infrared imaging guidance detector’s tracking recognition algorithm. The effect of laser dazzling varies depending on the position of the target object. Specifically, when the target object is at a farther distance, a higher laser power is required to successfully interfere with the infrared imaging detector. Conversely, when the target object is at a closer distance, only a lower laser power is needed to achieve dazzling. This finding highlights the importance of distance in laser dazzling. Due to beam scattering and attenuation, a higher laser power is required to maintain a sufficient intensity for dazzling when the target is farther away. In contrast, when the target is closer, the beam’s energy is more easily focused on the target; thus, a lower laser power can effectively interfere with the target.

To further refine the evaluation of the laser dazzling image, considering the impact of the laser power on the dazzling results, the normalized correlation between the template image and the dazzling image (Formula (6)) was used to calculate the evaluation value. The relationship between the maximum normalized correlation and the target position and laser dazzling power is shown in Figure 6.

Based on the recognition results and the correlation evaluation curves for different laser powers, combined with the human visual system (HVS), the following conclusions can be drawn:For targets at a close distance (r = 1.1 m), select the dazzling correlation threshold lgN_01 = −0.331 (position marked with ☆ on the curve in Figure 7), i.e., N_01_ = 0.467. Therefore, dazzling can be considered successful when the normalized maximum correlation N ≤ 0.467.For targets at a long distance (r = 1.7 m), select the dazzling correlation threshold lgN_02 = −0.226 (position marked with ☆ on the curve in Figure 7), i.e., N_02_ = 0.542. Therefore, dazzling can be considered successful when the normalized maximum correlation N ≤ 0.542.

These thresholds were selected based on the analysis of the recognition result images and the judgment of the human visual system. By analyzing the normalized maximum correlation, we determined the conditions under which dazzling can be considered successful. Thus, the evaluation system $F\left(P,r\right)$ was obtained, which included laser power lgP and target position lgr evaluation factors, and is defined as:(7)$$F\left(P,r\right)=AlgP+\left(1-A\right)lgr$$
where P is the laser power and r is the distance between the target and the laser center. $F\left(P,r\right)$ represents the normalized correlation function of the laser dazzling images under different target positions and different laser power conditions. This function comprehensively evaluates and judges the target tracking and recognition algorithm under different dazzling distances and laser powers, and linearly fits the target position distance and laser power evaluation factor. Coefficient A is used to adjust the weight of the target distance. The smaller the value of A, the greater the weight of the target position distance. From the analysis of the experimental results, the evaluation formula is in agreement with the experimental results.

### 5.2. Experimental Results and Analysis of the Laser Damage Effect

To quantify the degree of image distortion in the infrared imaging system before and after laser dazzling in the presence of functional damage, experiments were conducted using the experimental setup shown in Figure 3 under far-field conditions with a 10.6 μm pulsed laser for damaging the long-wave non-cooled camera. The camera’s built-in video recording function was used to record the imaging situation before and after laser irradiation. The partial damage images obtained from the damage effect experiment are shown in Figure 7.

The camera material was vanadium oxide, as confirmed by the laser damage images. To calculate the damage threshold of the camera under different irradiation times, it was necessary to obtain the value of the power reaching the target surface of the camera and the spot size. Camera lens 1 and lens 3 in front of the power meter probe were placed parallelly and adjacently, respectively, with the same F-number but different apertures. Lens 1 had a diameter of 50 mm, and lens 3 had a diameter of 40 mm, resulting in a diameter ratio of 1.25 and a light transmission ratio of the square of the diameter ratio, which was 1.5625. Based on this ratio, the power reaching the target surface after lens 1 can be calculated.

The spot size of the laser at a distance of 2 km was assessed using a long-wave monitoring camera, with a spot diameter of approximately 1.8 m. Therefore, the actual divergence angle of the laser in the far field was approximately 0.9 mrad. The focal length of the camera lens was 50 mm, resulting in a spot diameter D of approximately 45 μm on the camera’s photosensitive surface. This allowed for the determination of the damage threshold of the camera under different irradiation times, as shown in Table 3.

This study only analyzed the situation when the camera suffered from functional damage. In the second experiment, camera 1 experienced functional damage, with a quarter of the screen displaying as completely black. This area was unable to produce any imaging and could not be recovered even after power cycling, indicating irreversible damage. Although the imaging in other areas remained normal, the contrast of the imaging decreased with an increase in the amount of damage. Additionally, the non-imaging area of the entire camera significantly expanded until it was completely blind, rendering the camera completely ineffective and unable to restart. Therefore, the energy density threshold required for the functional damage of the camera was approximately 3 J/cm^2^.

The images in Figure 8a,b show different positions of the target image, with a size of 30 × 30 mm, selected from the background before the damage.

Due to the partial visibility of the target image in the presence of functional damage effects on the camera, an investigation into the dazzling assessment of the detector when functional damage occurs was conducted. The experimentally obtained damaged images were processed using the contour curvature-based Canny algorithm and the cross-correlation template matching algorithm, as described above. The backgrounds selected from Figure 8a,b at both far and near distances were used as the targets. The relationship between the laser energy and normalized correlation is depicted in the following figure.

The figure illustrates the correlation between the laser energy and normalized correlation, providing insights into the dazzling assessment effect of the detector under functional damage conditions.

Based on the correlation evaluation curves of damaged images and different laser energies, combined with the human visual system (HVS), for targets at a close distance, a correlation threshold of N_03_ = 0.449 was selected (marked as □ in the near-distance correlation evaluation curve in Figure 9). For targets at a far distance, a correlation threshold of N_04_ = 0.481 was chosen (marked as □ in the far-distance correlation evaluation curve in Figure 9). It is observed that the dazzling assessment conclusions are equally applicable to evaluate the damage results. At different distances, varying laser energies are required for successful dazzling with the recognition algorithm of the infrared imaging detector. A higher laser energy is needed for successful dazzling at a far distance, while a lower laser energy is sufficient for dazzling at a close distance. Additionally, the normalized correlation value at a far distance is slightly larger than the case of the target at a close distance, which is aligned with the human eye recognition results.

The correlation threshold refers to the threshold used in image processing to evaluate the similarity between the interfered image and the original, undisturbed image. This threshold is typically utilized to measure the extent of the quality loss in the image caused by laser irradiation. When the correlation between the damaged image and the original image falls below a certain set threshold, it indicates a significant deterioration in the image quality. In this study, when the correlation between the interfered image and the original image was below the set threshold, it signified the successful dazzling of the laser with the infrared imaging-guided detector resulting in tracking and recognition failure.

## 6. Conclusions

This study utilized the Canny algorithm based on contour curvature and the cross-correlation template matching algorithm to process and recognize 10.6 μm long-wave infrared laser dazzling images and damage images. Based on the maximum normalized correlation of the images, this paper proposed an evaluation system that can predict the effect of laser dazzling under different target positions and laser power conditions. The dazzling experiment results indicate that, at a far distance (r = 1.7 m), with a laser power P ≥ 1.5 mW, and normalized correlation N ≤ 0.542, recognition fails. At a close distance (r = 1.1 m), with a laser power P ≥ 0.4 mW, and normalized correlation N ≤ 0.467, dazzling is successful, leading to the failure of the infrared imaging-guided detector tracking and recognition algorithm. According to the damage effect experiment, the energy threshold for the functional damage of the long-wave non-cooled camera was determined to be 3 J/cm^2^. For targets at a close distance, a correlation threshold N_03_ = 0.449 was chosen, while for targets at a far distance, a correlation threshold N_04_ = 0.481 was selected. Based on the evaluation system proposed in this article, we can quickly determine whether laser dazzling is successful and assess its effectiveness. The discussion in this article focused on the issue of high-repetition-rate pulsed laser dazzling, which is not significantly different from continuous laser dazzling. Additionally, the evaluation system calculates the correlation based on the number of saturated pixels in the image. Since continuous laser dazzling also affects the number of saturated pixels on the detector, the same evaluation system is applicable to continuous lasers. This research holds significant military value and practical significance, laying the theoretical and experimental foundation for evaluating the operational effectiveness of infrared electro-optical countermeasure equipment and providing an effective and reliable quantitative evaluation basis for identifying and assessing dazzling equipment.

## Figures and Tables

**Figure 1 sensors-24-01827-f001:**
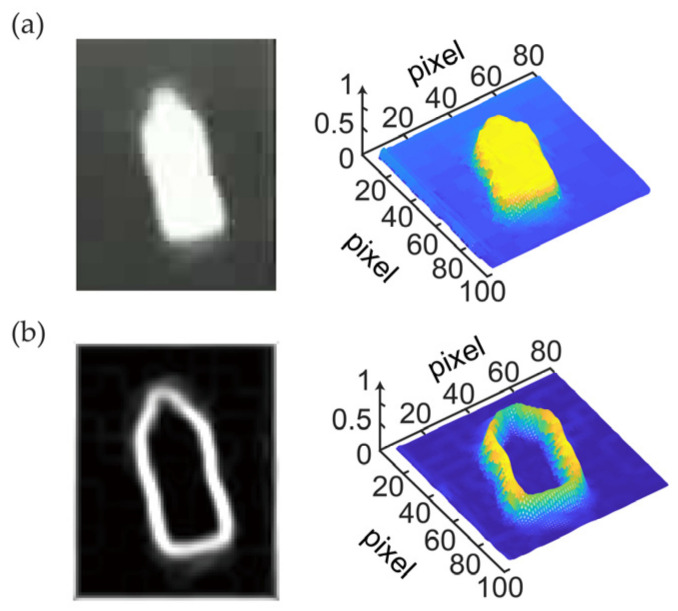
Image transformation and 3D stereogram. (**a**) Target image and its 3D intensity map; (**b**) image and its 3D intensity map processed by the Canny algorithm based on contour curvature. The color variation in the 3D intensity plot represents the normalized intensity of the image, with blue indicating lower intensity and yellow representing higher intensity. In other words, the transition from blue to yellow signifies the increase in intensity from low to high within the image.

**Figure 2 sensors-24-01827-f002:**
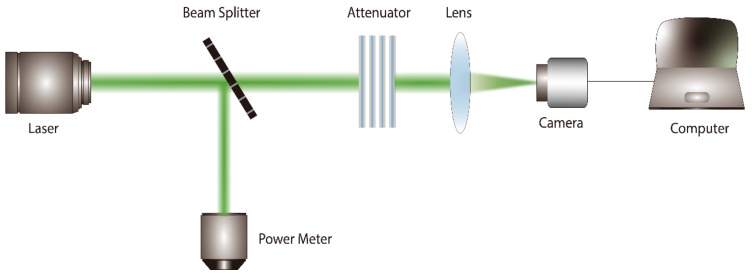
Dazzling effect experimental device concept diagram.

**Figure 3 sensors-24-01827-f003:**
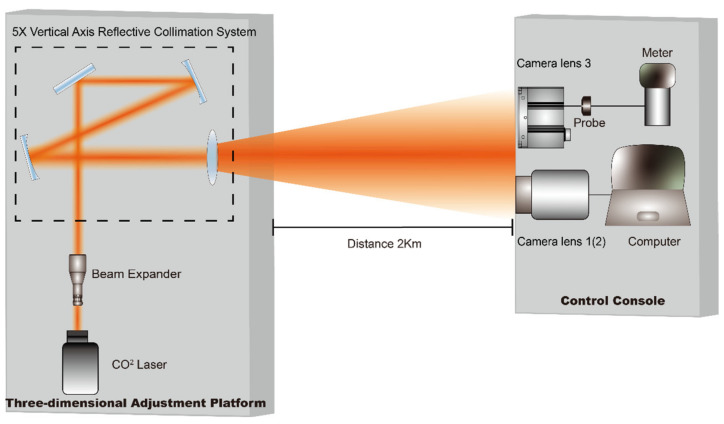
Schematic diagram of the remote damage experiment in an external field.

**Figure 4 sensors-24-01827-f004:**
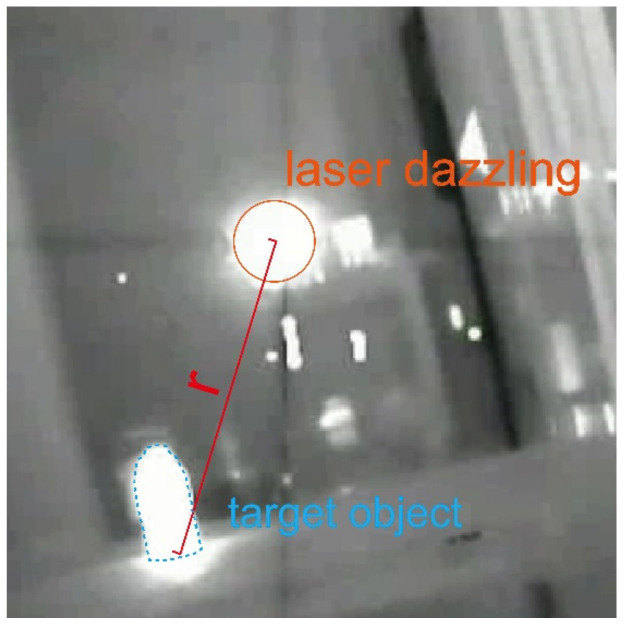
Infrared laser dazzling and target image.

**Figure 5 sensors-24-01827-f005:**
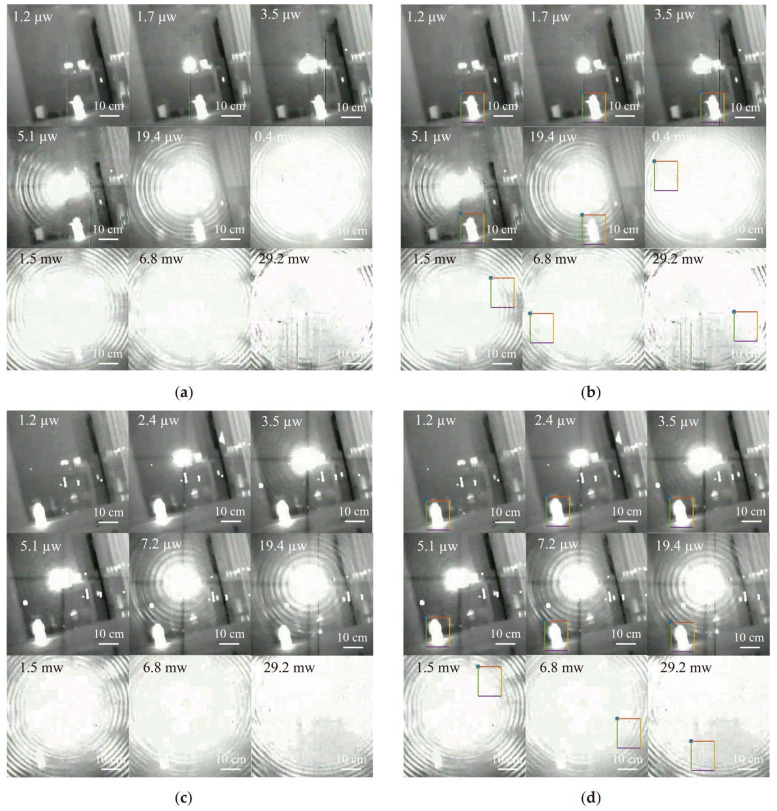
(**a**) Close-distance (r = 1.1 m) laser dazzling image. (**b**) Close-distance (r = 1.1 m) recognition of the dazzling images. (**c**) Long-distance (r = 1.7 m) laser dazzling image. (**d**) Long-distance (r = 1.7 m) recognition of the dazzling images. Figure 5b,d show the recognition results after algorithm processing, where the images inside the colored squares are the recognized images judged by the algorithm.

**Figure 6 sensors-24-01827-f006:**
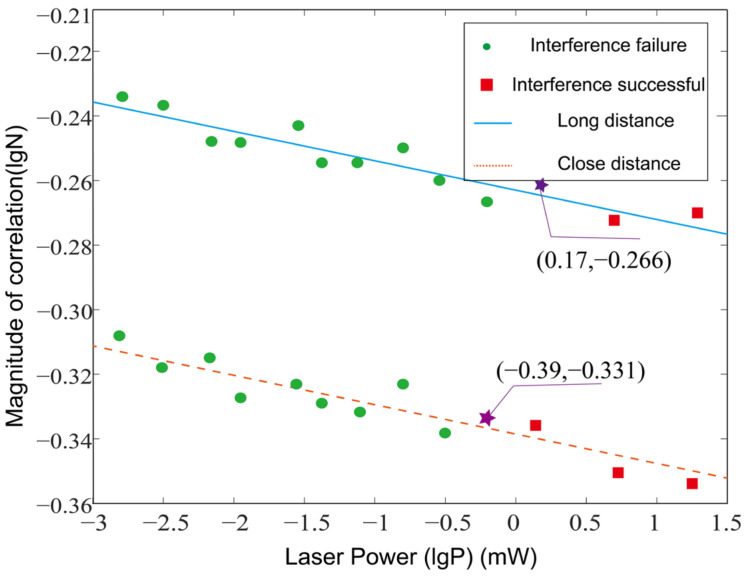
Different laser power similarity evaluation curves.

**Figure 7 sensors-24-01827-f007:**
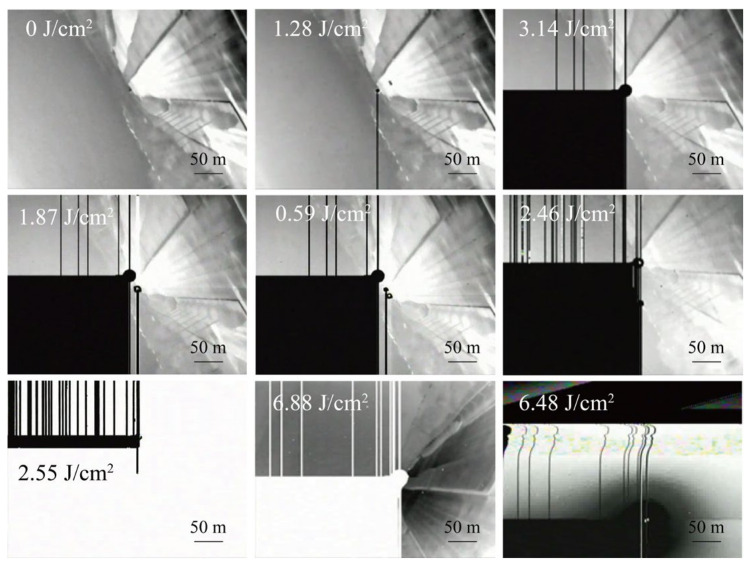
Damage morphology of different laser energy densities.

**Figure 8 sensors-24-01827-f008:**
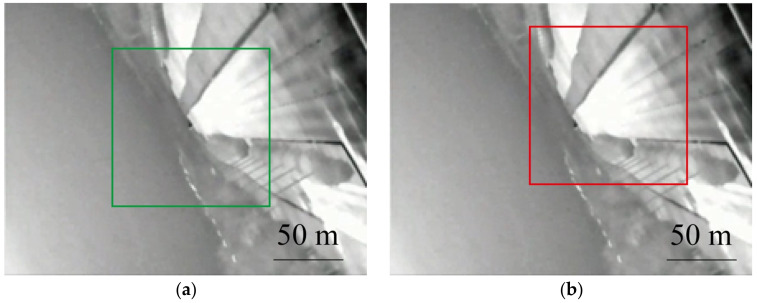
Different target images at different positions. (**a**) The green box represents the close distance target image. (**b**)The red box represents the long-distance target image.

**Figure 9 sensors-24-01827-f009:**
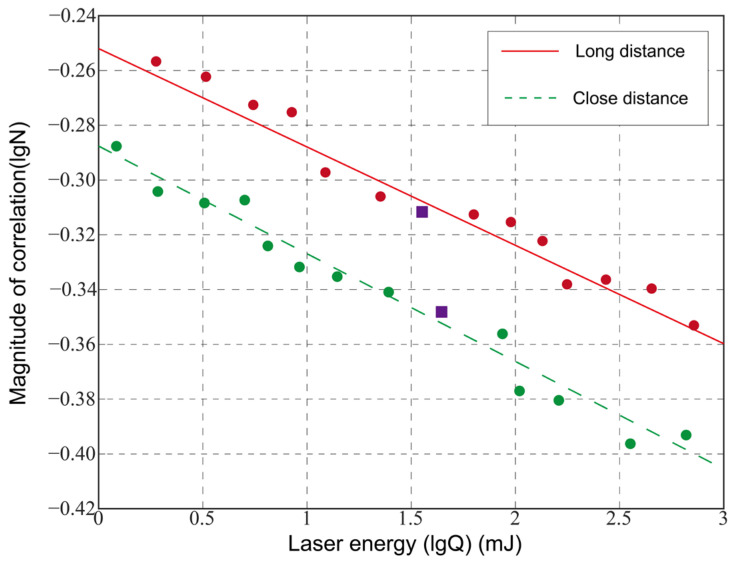
Evaluation curve of the different laser energy correlations.

**Table 1 sensors-24-01827-t001:** Transmittance of the different types of attenuators.

Attenuator Type	Transmittance
A Attenuator (5 pieces)	0.695
B Attenuator (2 pieces)	0.422
C Attenuator (4 pieces)	0.374
D Attenuator (5 pieces)	0.234

**Table 2 sensors-24-01827-t002:** Experimental measurement results of the laser dazzling effect.

Number	Power at Target (mw)	Spot Radius (μm)	Power Density at Target (W/cm^2^)	Number of Dazzling Pixels	
				Close Distance	Long Distance
1	0.001182	6.5	0.891310	10,413	10,428
2	0.001701	6.5	1.281858	10,938	10,992
3	0.002446	6.5	1.843532	11,726	11,842
4	0.003517	6.5	2.651317	12,197	12,301
5	0.005058	6.5	3.813050	15,271	15,362
6	0.007275	6.5	5.483823	15,587	15,693
7	0.019460	6.5	14.66922	26,081	27,947
8	0.052057	6.5	39.24018	41,897	42,064
9	0.139255	6.5	104.9674	62,527	64,691
10	0.372507	6.5	280.7880	73,856	74,119
11	1.594331	6.5	1201.772	74,798	75,586
12	6.823740	6.5	5143.587	95,052	95,986
13	29.20560	6.5	22,014.55	123,792	127,443

**Table 3 sensors-24-01827-t003:** Laser working conditions of the damage detectors.

Camera Number	Number	Irradiation Time/s	Repetition Frequency/Hz	Lens 1 to Target Power/mW	Spot Radius/μm	Energy Density (J/cm^2^)
Camera 1	1	1	100	2.03	22.5	1.28
	2	1	50	2.50	22.5	3.14
	3	1	100	2.97	22.5	1.87
	4	2	100	0.94	22.5	0.59
	5	1	100	4.84	22.5	3.05
	6	2	100	3.75	22.5	2.36
	7	3	100	3.91	22.5	2.46
	8	2	50	2.03	22.5	2.55
Camera 2	9	3	50	7.81	22.5	9.82
	10	4	50	5.47	22.5	6.88
	11	5	50	5.16	22.5	6.48

## Data Availability

The data presented in this study are available upon request from the corresponding author.

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
