# Peer review of "Dazzling Evaluation of the Impact of a High-Repetition-Rate CO2 Pulsed Laser on Infrared Imaging Systems"

_sensors, 2024, doi:10.3390/s24061827_

Round 1

Reviewer 1 Report

Comments and Suggestions for Authors

The main part of the manuscript is given as to damaged images

quote

The camera experienced functional damage, with a quarter of the image displaying 324 as completely black. This region is entirely non-imaging, and reconnecting after power- 325 off does not restore functionality, indicating irreversible damage. While other areas of the 326 imaging frame remain normal, the contrast of the imaging frame decreases with an in- 327 crease in the number of damages. Moreover, the non-imaging area of the entire camera 328 also expands significantly until blindness occurs, leading to complete camera failure. Re- 329 starting the camera does not restore functionality. Therefore, the energy density threshold 330 required for functional damage in the camera is approximately 3 J/cm².

end of quote

This is clearly stated. Now here is my problem

quote

Based on the correlation evaluation curves of damaged images and different laser 348 energies, combined with the Human Visual System (HVS), for targets at a close distance, 349 a correlation threshold of N03=0.449 is selected (marked as â–¡ in the near-distance correla- 350 tion evaluation curve in Figure 9). For targets at a far distance, a correlation threshold of 351 N04=0.481 is chosen (marked as â–¡ in the far-distance correlation evaluation curve in Figure 352 9). It is observed that the dazzling assessment conclusions are equally applicable to evaluating the damage results. At different distances, varying laser energies are required for 354 successful dazzling with the recognition algorithm of infrared imaging-guided weapons. 355 Higher laser energy is needed for successful dazzling at a far distance, while lower laser 356 energy is sufficient for dazzling at a close distance. Additionally, the normalized correla- 357 tion value at a far distance is slightly larger than the case of the target at a close distance, 358 which aligns with human eye recognition results

end of quote

The problem is, in explaining the following terminology

quote

Higher laser energy is needed for successful dazzling at a far distance, while lower laser 356 energy is sufficient for dazzling at a close distance. Additionally, the normalized correlation value at a far distance is slightly larger than the case of the target at a close distance, 358 which aligns with human eye recognition results]\

end of quote

 While this may be fully accessible to specialists, general laser physics people and engineers simply will NOT understand this

Finally in the conclusion

quote

For targets at a close 371 distance, a correlation threshold N03=0.449 is chosen, while for targets at a far distance, a 372 correlation threshold N04=0.481 is selected. Based on the proposed evaluation system, it is 373 Sensors. 2024, 16, x FOR PEER REVIEW 12 of 13 possible to quickly determine the success of laser dazzling and evaluate its effectiveness. 374 The discussed evaluation system is applicable to both high-repetition-rate pulsed laser 375 dazzling and continuous laser dazzling, providing a quantitative assessment standard for 376 the impact of laser active imaging systems on target recognition capabilities

end of quote

I recognize some of what is said here. I was present when my wife got a PhD in laser physics so it is not Greek to me. However

I doubt many would. It is mandatory that there be expansion of this last section with some reasons given as to the choices given in this last quoted section. Above all this requires explanation'

quote

At a close distance (r=1.1 m), with 367 laser power P≥0.4 mW and normalized correlation N≤0.467, dazzling is successful, leading 368 to the failure of the infrared imaging-guided weapon tracking and recognition algorithm. 369 According to the damage effect experiment, the energy threshold for functional damage 370 to the long-wave non-cooled camera is determined to be 3 J/cm2 . For targets at a close 371 distance, a correlation threshold N03=0.449 is chosen, while for targets at a far distance, a 372 correlation threshold N04=0.481 is selected.

end of quote

This needs context. Otherwise it is  jargon which has no clear reference point outside of a lab

I am familiar with the term correlation threshold. I doubt few people are. As this is very important to the wrap up of this document a paragraph describing its relevance to the conclusion of this project should be included

Reviewer 2 Report

Comments and Suggestions for Authors

In the present article authors report dazzling effect of high repetition rate CO2 pulsed laser on infrared imaging system by using Canny edge extraction algorithm. After going through the manuscript I found that it lacks and needs comprehensive clarification and revision.

Points of concern are

A detailed and critical analysis of the presented figures can improve the quality of the manuscript.

Experimental details are not clear e.g. beam splitter, attenuator, camera used. How image is formed? How the attenuator controls the irradiation time? What is the pulse width of the laser? What is the spot size? If it is 6.5 µm for all the cases? How power density is estimated? If the beam has Gaussian shape?

Details of the experimental setup (Fig. 4) need to be given. How 2X and 5x coaxial expansions are achieved? Where is lens L2 in the mentioned schematic diagram? Output of camera 1 is shown to be fed to the computer but the output of camera2 has only link with energy meter.

A is missing in Fig. 3.

Introduction can be shortened by mentioning saline aspects of the relevant works.

In Fig. 6, why there is negative power?

Eq.7, what is the rationale behind considering coefficient A for lgP and (1-A) for lgr.

L278-279, sentence appears rather vague.

L330, How non-functioning of the camera can be considered as damage threshold? It should depend on the material of the sensor of the camera.

Comments on the Quality of English Language

English is fine excepting a few minor issues.

Round 2

Reviewer 2 Report

Comments and Suggestions for Authors

Authors have revised it considerably according to the reviewers' suggestions. However, these minor issues have to addressed before it is finally accepted.

(i) L 37, Abstract, Laser suppression dazzling does not appear to be proper. It should be replaced by suppression of laser dazzling

(ii) L 217, what do you mean by success rate meter?

(iii) L221, 224 define P, f1 and f2.

(iv)L248, respectively should be after expander.

Comments on the Quality of English Language

Minor editing of English is required.
